# Prospective Feasibility and Revalidation of the Equine Acute Abdominal Pain Scale (EAAPS) in Clinical Cases of Colic in Horses

**DOI:** 10.3390/ani10122242

**Published:** 2020-11-29

**Authors:** Yamit Maskato, Alexandra H. A. Dugdale, Ellen R. Singer, Gal Kelmer, Gila A. Sutton

**Affiliations:** 1Large Animal Department, Robert H Smith, Faculty of Agriculture, Food and Environmental Sciences, Koret School of Veterinary Medicine, Veterinary Teaching Hospital, The Hebrew University of Jerusalem, P.O. Box 12, Rehovot 7610001, Israel; yamit.maskato@mail.huji.ac.il (Y.M.); gal.kelmer@mail.huji.ac.il (G.K.); 2Leahurst Campus, School of Veterinary Science, University of Liverpool, Chester High Road, Neston CH64 7TE, UK; alexdugdale@btinternet.com (A.H.A.D.); singer844@btinternet.com (E.R.S.)

**Keywords:** pain assessment, visual analogue scale, validity, usability, utility

## Abstract

**Simple Summary:**

Severity of pain, an important parameter in clinical decision making, is subjective. The Equine Acute Abdominal Pain Scale (EAAPS) was evaluated for the first time in 237 horses presenting with colic (abdominal pain) at two equine hospitals; in Israel and in the United Kingdom. The EAAPS demonstrated validity and was reportedly quick and easy to use. The EAAPS is the only equine pain scale that has been tested to this extent for these properties on clinical cases in equine hospitals. Use of the EAAPS apparently requires no training, is easy to use in clinical cases, and can improve equine welfare.

**Abstract:**

Assessment of the severity of pain in colic cases is subjective. The Equine Acute Abdominal Pain Scale (EAAPS), previously validated using film clips of horses with colic, was tested for feasibility and revalidated in both medical and surgical colic cases in Israel and the UK. Feasibility qualities evaluated were quickness and ease-of-use. Pain in 231 horses, presented for colic, was assessed by 35 participants; 26 in the UK and 9 in Israel. Without prior training, participants assessed the severity of pain using two scales; the EAAPS and a visual analogue scale (VAS). Convergent validity comparing the EAAPS to the VAS was substantial, discriminant validity was good, and predictive validity for surgical treatment was similar to the VAS, but for mortality, the VAS was significantly better. No participants reported the EAAPS to be “very slow” or “very difficult” to use. The mode reported was “quick”/“very quick” and “easy”/“very easy” to use, though in less than 10% of cases, it was reported to be a little less quick or easy. More experienced first-time users found it significantly quicker to use than less experienced participants. In conclusion, the EAAPS is the only equine pain assessment scale that has been tested and found to demonstrate good feasibility for use in the referral hospital setting.

## 1. Introduction

Colic is a common condition of horses characterised by abdominal pain and high mortality. In 2015, it caused 31% of deaths in horses aged 1 to 20 years, making it the most common cause of equine death in the USA [1]. Research and clinical experience have shown that severity of pain is an important parameter for clinical decision making, monitoring patient status, evaluating analgesia effectiveness, humane end-point decisions, as well as for research [2,3]. However, pain assessment is subjective [4]; thus, when global rating scales, such as numerical rating scales (NRS) and visual analogue scales (VAS), are used, experience with horses [5] and interpretation of pain depends on understanding the species’ normal behaviour. A major difficulty with global rating scales, lacking any descriptive criteria for pain assessment, is that they may be subject to many observer biases [6]. They are also not very reliable [7,8], and in repeated measurements, measurement error causes inconsistent results [9].

There have been many attempts to devise pain scoring systems in horses [10,11]. Less subjective scoring systems, such as composite pain scales incorporating physiological and behavioural parameters, demonstrate higher interobserver reliability compared to VAS [12]. Physiological variables, however, are not specific for pain; thus, behaviours, induced by pain, have become a focus of pain assessment [13].

Among the pain scales based on behaviours, the Horse Grimace Scale and the equine facial expression pain scale require training [14]. The Equine Acute Abdominal Pain Scale (EAAPS) utilises typical colic behaviours with good to excellent agreement and minimal bias between observers (equine veterinarians) viewing films of horses with colic [15]. Therefore, training may not be needed for observers with experience in treating colic cases.

The EAAPS is a simple descriptive scale. A simple one-digit score is given to grade the severity of pain based on descriptors to guide the choice of the score (Table 1). The EAAPS has been shown to have significantly higher interobserver and intraobserver reliabilities than either global scale, VAS or NRS or another simple descriptive scale [8,15,16,17]. These results indicate that EAAPS can be relied upon due to its reproducibility [10,13,14].

Reliability is a necessary aspect of subjective health measurement scales since it indicates to what extent the scale can be relied on to be consistent; however, the scale must also be valid. Validity conveys how much the scale measures what it aims to measure. Ideally, validity involves comparison with a gold standard (criterion validity). When there is no gold standard, as for the measurement of pain, alternate types of validity are required, and for that reason, validity needs to be reassessed under new situations that arise. Alternate types of validity include construct validity. A construct is a mini-theory. In this case, the mini-theory is that pain would, on the whole, be more severe in certain situations, and therefore, a valid pain assessment tool would be expected to reflect that theory by obtaining higher scores in those situations. Discriminant and predictive validities are construct validities. Discriminant validity means that the scale can discriminate between extreme groups of pain (mild or no pain compared to severe pain). Predictive validity means that higher scores achieved with the scale will be significantly associated with more severe colic, such as surgically-treated, in comparison with less severe medically-treated cases, or cases that died compared to cases that lived. An additional validity, convergent validity, is one that shows a correlation between the new scale and a more established scale, such as the VAS [9]. In this way, the VAS, although still subjective, could be used as a “gold standard.”

Up until now, film clips of horses with colic have been used for assessment of the EAAPS rather than cases presenting in the clinical setting, which is the intended use for the EAAPS. When designing a subjective measurement scale for clinical use, the feasibility of the scale is important to assess [18,19,20]. The concept of feasibility is that the scale should be appropriate for use in clinical practice [19]. The desirable feasibility qualities include ease of learning and efficiency or how quick it is to use [21] as well as acceptability by users [22] and other properties, such as cost, intrusiveness, degree of error, consequences of errors, and so on. The semantics of the concept of feasibility are not uniform. In addition to the term feasibility, the terminology includes utility, usefulness, usability, and more [9,21,22], although the actual properties they relate to are quite similar. For the remainder of this paper, the term feasibility will be used. Feasibility studies for pain scales in the medical literature commonly evaluate only a few of the potential aspects [21,22].

The aim of this study was to evaluate the use of the EAAPS by hospital staff in real time. The primary focus was on the feasibility of the scale. The two particular aspects of feasibility to be investigated in the current study were the time investment required and the complexity of the EAAPS. These aspects are important since the EAAPS was to be used in emergency cases where time and simplicity are important features for a pain scale. The main hypothesis was, therefore, that the EAAPS would be quick and easy to use. However, since the EAAPS was being assessed prospectively for the first time on clinical cases in this study, validity was evaluated for the first time under real-time conditions. Convergent, discriminant, and predictive validity were examined in this study, in addition to the main focus, which was how easy or quick it was to use the EAAPS in the clinical emergency situation.

## 2. Materials and Methods

### 2.1. Study Design

This was a prospective descriptive study carried out in two institutions. Participants included veterinarians, technicians, and veterinary students at the Koret School of Veterinary Medicine Teaching Hospital (Israel) and at the Philip Leverhulme Equine Hospital, University of Liverpool (UK). Participants evaluated horses older than 1 year of age that presented with acute abdominal pain or colic. Pain severity was assessed by using both a global VAS and the EAAPS. The EAAPS was evaluated for three types of validity: convergent, discriminant, and predictive. Feasibility qualities of quickness and ease-of-use were assessed subjectively by endorsement by the participants.

### 2.2. Questionnaires

The questionnaires used by the participants included information regarding the horses (age, breed, sex, heart rate on admission to the hospital, surgical or medical treatment, mortality), and characteristics of the participants (age, education, and prior experience with the EAAPS). The questionnaire had a VAS and an EAAPS. The VAS was a 10 cm line with the statement “pain free” at the left end of the line and “severe pain” at the right end. The written instructions were to mark a cross on the line so that the distance from the “pain free” side represented the pain score. The EAAPS had six categories of pain based on weighted behaviours. The participants were instructed to circle the behaviours that the horse was demonstrating and to choose the pain score of the behaviour farthest to the right. The scores of the EAAPS were given as letters (zero, A, B, C, D, E) rather than numbers (zero to 5) to reduce the bias of intuiting the significance of the score chosen. The scores were presented from left to right. The participants were asked to score the severity of pain with the VAS before using the EAAPS so as not to be biased by the categories of pain assigned to the behaviours in the EAAPS, particularly when using the EAAPS for the first time. The staff was not trained to use either the VAS or the EAAPS other than the written instructions on the forms. The questionnaires also contained questions relating to the feasibility of the EAAPS (quickness and ease-of-use), and there was an additional open-answer optional question regarding the satisfaction with the EAAPS (Table 1). The forms used in Israel were independent of the medical record; however, in the UK, the information was either embedded (VAS) or combined (EAAPS) with the medical record forms normally used on admission of horses to the hospital (Appendix A).

### 2.3. Validation

The validation methods were previously described [8] except for the following. Convergent validity was assessed as agreement beyond chance (kappa) between the EAAPS score and the VAS score, which was used as a “gold standard.” Spearman correlations were also calculated between the heart rate and the two scales as a form of convergent validity.

Discriminant validity was expressed as the ability of the EAAPS scores to discriminate between two equally sized extreme groups made by using the median of the VAS scores over all of the cases. Horses with VAS scores above the median were considered to be demonstrating severe pain, and those below were considered to be demonstrating mild pain. Receiver operating characteristic (ROC) curves were then used to assess the association between the scores given by the EAAPS and the group demonstrating severe pain.

As previously described, predictive validity was assessed as the scale’s ability to demonstrate an association with or “predict” mortality or surgical intervention, based on the construct that severe pain reflects severe illness of the horse and that severe illness involves a higher probability of surgical treatment and a higher mortality rate [23]. Predictive validity of the EAAPS and VAS to predict mortality and surgical intervention by ROC curves were compared. Likelihood ratios of a positive test (LR^+^), calculated as (sensitivity/1-specificity), for the EAAPS and VAS to predict mortality and surgical treatment were based on the sensitivity and specificity from the ROC curve to predict mortality or surgical treatment. The cut-off value chosen for each scale was the point associated with the highest LR^+^ calculated.

### 2.4. Feasibility

Qualities of feasibility of the EAAPS were evaluated by the participant’s rank of the quickness (very quick/quick/within reasonable time/slow/very slow) and ease-of-use (very easy/easy/not easy–not difficult/difficult/very difficult) of using the EAAPS. Feasibility of the EAAPS was expressed as the percentage of replies in each category over all of the completed forms for first-time users only. When evaluating for a learning curve or change in response from the first-time use to when the scale had been used multiple times, only those using the scale for more than three times were included. In those cases, the most frequent response of each participant (mode) was chosen from the fourth time used until the last time used. This was done to reduce the bias of clustering produced by participants scoring varying numbers of horses.

### 2.5. Statistical Analysis

Statistical analysis was made using IBM SPSS 18, 22, 24, IBM, USA, unless otherwise noted. Ordinal parameters were tested for normal distribution by Shapiro–Wilk normality test. Categorical parameters were presented as frequencies and percentages.

Convergent validity of the EAAPS in comparison to the global VAS was expressed by quadratic weighted kappas with the 95% confidence interval of the score each horse was assigned using each scale. For this calculation, the Vassar Stats online kappa calculator was used [24]. For the values of the VAS to correspond to the categories of the EAAPS, the VAS scores were mathematically converted to a round number between 0 and 5 by dividing the scores by 20 and rounding them to the closest integer. This value was compared to the EAAPS score of the same horse. Interpretation of the kappa coefficient was based on Landis and Koch [25].

For predictive validity, the association between the scale and the outcome of mortality or surgical treatment were tested by the Mann–Whitney U Test for the VAS and by Fisher’s Exact Test for the EAAPS. ROC curves were used to assess both predictive validities and extreme group validity. The areas under the curve (AUC) with 95% confidence intervals are reported. The scales were also assessed for statistically significant differences by comparing the AUC of one scale to the 95% confidence interval of the other. If the value of one was outside the 95% confidence interval of the other, the difference was considered significant at an alpha of 0.05. For expression of the strength of the effect, the LR^+^ were calculated for each ROC point or coordinate, and the LR^+^ with the highest values are presented with their associated cut-off values.

Results regarding the feasibility qualities were compared by Fisher’s Exact Test between participants of different age groups when using the EAAPS for the first time. The age of the participants was categorised into 4 groups: under or equal to 25 years old, 26 to 29 years old, 30 to 34 years old and 35 years or older.

## 3. Results

A total of 243 forms were collected in the study: 218 forms in the UK and 19 forms in Israel. Six of the forms from the UK were excluded from the study (2.5% of the study population) due to horse age (<1 year) and species (donkey) that did not fit the inclusion criteria. The remaining 237 forms were included in the study. Some of the forms from Israel were from the same five horses; thus, there were 13 horses for 19 forms.

### 3.1. Missing Data

There was missing information regarding the characteristics of the participants (Table 2) and 5% missing data relating to the horse population (Table 3). Values of heart rate and pain scores by the EAAPS and by VAS were not available for 6.3%, 25.7%, and 17.7% of the horses, respectively. Only those cases in which the EAAPS and the VAS were both available were included in the assessment of agreement between them (convergent validity). Otherwise, missing data were treated as missing data by default of the software programme (IBM SPSS).

### 3.2. Population Characteristics of Participants

There were a total of 35 participants: 26 from the UK and 9 from Israel. The age range was 22–55 years old. The majority of participants in Israel were students (56%). This information was not available regarding the UK participants; however, the range of ages in the UK was wider. Whereas in Israel, the ages ranged from 26 to 35 years old, 35% of participants in the UK were under 25 years of age and 15% were greater than 36 years old (Table 2). Participants in the UK evaluated more horses than participants in Israel. The ratio between the number of participants to the number of horses evaluated in Israel was 1:2, whereas in the UK, the ratio was 1:6. Eight participants in Israel and 11 in the UK graded the EAAPS only once, 1 in Israel, and 9 in the UK graded over 3 times. In the UK only, eight graded over four times and five over five times. One participant in the UK graded pain with the EAAPS 41 times and registered their opinion each time.

### 3.3. Horse Population

Of the 231 horses included in the study, the most common breed in Israel was the Arabian horse (46%), whereas in the UK, about a fifth of the horses were Welsh or Cob breeds (22%), the most commonly represented group in the study overall. The breeds, ages, and sexes of the horses were categorised into groups (Table 3). The ages ranged from 1 to 34 years. The median heart rate was 48 beats per minute (bpm) with the interquartile range (IQR) (40, 60) and absolute range of 36–80 bpm. The median pain score of the cases, based on the global scale (VAS), was 25 out of 100 mm, with an IQR of (12, 64).

### 3.4. Clinical Information

Treatment modality is known for 228 of the 231 cases (99%). Medical treatment was chosen for 99 (43%) cases and surgical treatment for 129 (57%). Out of 225 cases for which the outcome is known (97%), 164 (71%) survived. Five horses presented with abdominal pain unrelated to the gastrointestinal tract (pneumonia and pleuritis, urolithiasis, ovarian torsion, and hemangiosarcoma). Euthanasia was performed in 58 cases (25%) for a variety of reasons, including economic constraints, poor prognosis, recurrence of clinical signs, inability to stand after surgery, grass sickness, and deteriorating enterocolitis. Three horses died: one eviscerated, one died in the immediate postoperative recovery period without standing, and one died following induction of general anaesthesia.

### 3.5. Validation Tests

The convergent validity of the EAAPS compared to the VAS (agreement) was substantial (quadratic weighted kappa = 0.64; 95% confidence interval (95% CI): 0.55–0.73). The correlation between the EAAPS and the VAS was a Spearman’s rho of 0.7; however, each of the scales correlated poorly with the heart rate (rho with a VAS of 0.2; rho with an EAAPS 0.003). The discriminant validity of the EAAPS was good, with an AUC of 0.85 and 95% CI: 0.76–0.93. At a cut-off of 2.5, the EAAPS had a likelihood ratio of a positive test of 6.4. Regarding predictive validity, higher EAAPS and VAS scores were significantly associated with surgical treatment and mortality (*p* < 0.01). The ability to predict the outcome of each scale can be seen in Table 4.

### 3.6. Feasibility

#### 3.6.1. First Time-Users

No participants using the EAAPS for the first time responded that the EAAPS was “very difficult” or “very slow” to use (Table 5). However, the older participants chose significantly more positive responses regarding the quickness of the scale than the younger participants (*p* < 0.01). There was no significant difference in regards to ease-of-use (Table 6).

#### 3.6.2. Learning Based on Experience

Only one of the participants in Israel and nine participants in the UK used the EAAPS scale more than three times. The participant in Israel was a student who responded that assessing the pain severity with the EAAPS was “slow” and “not easy/not difficult” the first time, but then, after using it more than three times, the responses were that it was “quick” and “easy.” Of the ten who used the EAAPS more than three times (one in Israel, nine in the UK), the mode response was “very quick” (6/10) or “quick” (3/10), and in one, responses were split equally between “quick” and “very quick.” For ease-of-use, the mode was “very easy” (4/10) or “easy” (5/10) and split between “very easy” and “easy” for 1/10.

For those who used the EAAPS more than three times, the response on the first time could be compared to the response after multiple uses (>3 times) only for seven participants (one in Israel and six in the UK). All of these seven participants were over 25 years of age. Although the mode of the responses changed more often in the direction of more quickly or easier, a significant change was not demonstrated.

#### 3.6.3. Exceptions

Four participants who used the EAAPS many times (>3 times) occasionally chose a response indicating that the EAAPS was not as quick or as easy to use as the mode would indicate. Regarding the feasibility quality of quickness-of-use of the scale, the response chosen was “slow” (2/99 horses) or “within reasonable time” (7/99 horses). Regarding ease-of-use of the scale, the response chosen was “difficult” (5/99 horses) or “not easy/not difficult” (8/99 horses).

#### 3.6.4. Open-Answer Optional Question

Written comments were few but included: the EAAPS was too restrictive or that the behaviour of the horse did not fit the descriptors. For example, there were comments that posturing to urinate was not in the pain scale. Other comments mentioned that shivering, trembling, muscle fasciculations, sweating, being quiet, dullness, and depression were noted, but these were missing from the EAAPS. There were comments indicating difficulty judging behaviours in the stocks or if the horse was stressed or not demonstrating overt pain behaviours.

## 4. Discussion

There have been a number of pain scales constructed and tested for horses in the past decade or so [13,14,25]; however, this is the first study testing the use of a scale for feasibility on a large number of clinical cases in real-time. No aspects of feasibility have been assessed for any of the pain scales being constructed and validated for use in horses until now. The only previous validation study in a refereed journal carried out on acute colic cases in real time was a study designed to validate the EQUUS-COMPASS and EQUUS-FAP scales: however, the issue of feasibility was not addressed in that study. That study was carried out on 25 cases and 25 controls that were chosen for inclusion in the study [26]. In the study described here, all horses admitted to the UK hospital were included in the study as the forms for the study were an integral part of the colic admission forms.

In this study, the EAAPS was found to have two important qualities of feasibility, which were that the scoring system was quick and easy to use. In human medicine, feasibility is considered an important characteristic of a pain scale. The best scale is considered one with good feasibility and reliability [27]. The EAAPS has now demonstrated both these qualities, albeit the reliability was not retested in this study. The feasibility was tested under real-life conditions when the horse demonstrating pain needed to be assessed quickly and easily to make rapid clinical decisions necessary in emergency situations. Feasibility of the EAAPS had until now only been assessed when assessors viewed horses demonstrating colic in films [17].

The fact that none of the assessors in this study received training in the use of the EAAPS prior to scoring the horses, and yet, no participants felt that it was very difficult or very slow to use, supports the claim that the EAAPS is very quick and easy to use [28]. Although not objectively measured in this study, the subjective findings support the claim that it expectedly takes a matter of minutes or less to score a horse with the EAAPS, whereas some of the scales take up to 5 min to reach a conclusion [14]. Most of the other scales available today require some degree of training prior to use, such as the composite pain scales [12,14] and the facial pain scales [11].

Although not trained in the use of the EAAPS, most of the participants in this study were familiar with colic behaviours in horses. Prior experience with colic behaviours may have made using the EAAPS even easier, since the older participants found it significantly quicker to use than the younger participants when using the EAAPS for the first time. It is likely, due to the fact that the study was carried out in academic institutions, that the older age group had more prior experience in pain assessment in cases of colic presenting to the hospital. In the future, to investigate the effect of training, people inexperienced with colic behaviours should assess the feasibility of the scale, possibly before and after training, to fully explore the effect of training on the use of the scale.

A learning curve based on experience was not demonstrated in this study, perhaps because of the small number of participants who used the scale more than three times and the fact that all of those who did were in the older, more experienced age groups. However, despite the difference in the age groups or experience, even those with less experience did not find the EAAPS to be very difficult or very slow to use, supporting the claim of the simplicity of the scale.

The written comments that the behaviour of the horse did not fit the descriptors included in the EAAPS raised a valid concern and may explain the exceptions in which observers occasionally found the EAAPS to be more difficult or slower to use. However, adding signs or behaviours that observers in the previous studies did not agree about having seen in films of horses with colic would lead to a reduction in reliability and could cause the EAAPS to lose its advantage over the global scales [8,15]. Reliability is a necessary prerequisite for validity [9]. A scale cannot be valid if it is not reliable, and the EAAPS is significantly more reliable than a global numerical rating scale [8,16] or another unnamed simple descriptive scale described by Mair and Smith [17]. To maintain reliability, behaviours that observers could not agree upon seeing clearly in colic cases were not included in the EAAPS [15]. However, to minimise the occasions in which the EAAPS would be unable to adequately assess the severity of pain due to the lack of some descriptors, the reported frequency that behaviours are observed in cases of colic was also taken into account when the EAAPS was constructed. Common behaviours were preferentially included in the EAAPS [15].

Some behaviours mentioned in the comments, such as not being included in the EAAPS, are controversial. Depression is clearly a sign of the severity of colic because depression was included in the gravity score by Grulke et al. [23], considered a systemic sign of the severity of illness [2], and has been shown to be associated with the need for surgery; however, as a sign of pain, there is great controversy [2]. Depression is mentioned in regards to differing and conflicting types of pain. Depression was described as being associated with prolonged acute pain [29], with long-term conditions [30], chronic pain [2], as well as general pain [11]. Depression was included in the original EAAPS as a sign of mild pain [8] despite the controversy and despite the fact that it did not obtain good agreement when tested for agreement and was endorsed by only one of six experts [15]. Ultimately, due to the great diversity in the opinions of experts as to whether it belonged in the category of mild pain or the category of severe pain [15], and whether it is really a sign of pain at all, it was removed from the revised version of the EAAPS [16], the one used in this study.

Regarding shivering/muscle fasciculations or trembling, there was some consensus on construction of the EAAPS that one of these should be included in a pain scale [15]; however, shivering did not demonstrate agreement at all (kappa of 0.07; 2% agreement; unpublished data). For this reason, shivering was not included in the EAAPS, although the fact that shivering and sweating were hard to distinguish in the films, may have contributed to its lack of agreement beyond chance [15].

Sweating is a sign that has appeared in equine pain scales since the first equine abdominal pain scale of Muir and Robertson [31]. This tradition continued with the following pain scales published [4,10,32,33,34]. However, since sweating is a physiological autonomic response [29] to moderate to severe pain, it is a systemic sign of the severity of colic associated with severe pain rather than a sign of abdominal pain itself. In fact, like the EAAPS, the more recent pain scales have not incorporated sweating as a sign of pain [30,35], although sweating remains a component of the EQUUS-COMPASS [12].

Weighting is important in pain scales [11], and the EAAPS has weighting based on the severity of pain each behaviour indicates. However, the EAAPS does not have gradations for each of the behaviours, which would allow the observer to give a higher score if the behaviour occurs more frequently or more violently [15]. Including gradations in the pain scale increases the subjectivity and reduces reliability. This was shown when the EAAPS-2 was compared to the EAAPS-1. The two scales were identical, except that the EAAPS-1 did not have gradations whereas the EAAPS-2 did. Since the presence or absence of gradations was the only difference between the two scales, the gradations clearly rendered the EAAPS-2 less reliable. Only the EAAPS-1 was significantly more reliable than the other three scales [8], giving it an important advantage. Since the EAAPS has no gradations, it may be more suitable in the hands of lay people who may not have the experience necessary to choose the most appropriate gradation. Moreover, calculating a final score with the EAAPS is rapid since there is no need to sum up the gradations.

The EAAPS is a simple descriptive scale in contrast to most equine abdominal pain scales that are multidimensional composite scales. Intuitively, multidimensional scales have greater face validity (appear to be more valid) than the EAAPS since they include additional indicators besides behaviours. However, despite this apparent advantage, multidimensional scales may actually not be more valid. For example, multidimensional scales may include physiological parameters such as heart rate; however, heart rate has not been found to be a reliable measure of pain in horses [5,11,29]. Although heart rate is easily and objectively measured and increases with severe pain, it is also affected by many factors other than pain, such as fever, excitement, anxiety, shock, endotoxaemia, and medication [29]. In this EAAPS study, there was a poor correlation between the heart rate and each of the pain scales, whereas there was a good correlation between the scales. Therefore, it may not be beneficial to include heart rate in a pain scale, as was found in pain assessment of infants and children [36] as well as in postoperative pain assessment of horses [37].

Multidimensional scales are usually composite scales using several simple descriptive scales with clearly defined grades chosen by the assessor and then summed up [11]. Therefore, although the composite pain scales may have greater face validity than the EAAPS, they take more time to apply [2] and may be less suitable in an emergency situation compared to the EAAPS, which provides an immediate score.

Clinically relevant interpretation of the scores achieved by multidimensional scales can be difficult [2]. For example, using Bussières et al.’s composite pain scale [34], despite the maximum possible score of 39, the median score among non-survivors, those expected to be more painful cases of colic, was around 10, with an interquartile range of around 5.15 or less [38]. Therefore, an inexperienced or untrained user of this composite pain scale may believe a score of 10 out of 39 is not very painful; however, in clinical trials, it was shown to be so. On the other hand, the EAAPS scale of 0–5 is intuitive and is apparently feasible, even with no prior training.

Blinding was not considered relevant in this study because the pain assessment was performed at the time of colic admission prior to a diagnosis or treatment plan being formed, compared to a pain scale used postoperatively, where the surgical findings are known and may influence the pain assessment [36].

Similarly to the previous EAAPS studies utilising film clips of horses [8,16,17], in this study, the EAAPS was found to be valid in cases in real time, supporting previous EAAPS studies utilising film clips of horses [8,16,17]. The EAAPS was comparable to the VAS for convergent validity and extreme group validity as well as for predictive validity regarding surgical treatment. The EAAPS was not as good as the VAS at predicting mortality. However, since pain severity is not always associated with mortality, it is possible that the VAS recognised other clinical signs, such as heart rate and autonomic responses, that indicate the gravity of the case and which may be stronger prognostic indicators than pain alone.

Limitations of the study include compliance with using both the EAAPS and the VAS. Although 99% of the participants scored the pain severity using one of the scales, only 53% used both the EAAPS and VAS, as directed. Reliability was not tested as this is difficult to do in acute colic cases due to the ethical concerns of negatively impacting the treatment of the horse by requiring more than one person to independently assess the horse’s pain at exactly the same time to calculate interobserver reliability. There were also concerns regarding further reducing compliance by making demands on clinicians who were not invested in the study, thereby reducing the sample size in a likely biased manner.

Two institutions were involved; however, due to the overwhelming difference in the sample sizes of the two institutions, the most significant results emerged from the UK. Information regarding the education of the UK participants was missing leading to the assumption that age represented experience with colic cases.

Another limitation of the study was that the participants were not instructed to use the EAAPS when the horses were unrestrained and free in a stall. This feature could impact the results since restrained (in stocks) horses might not demonstrate pain behaviours as readily as when unrestrained. There was a conscious effort made to minimise the restrictions on the participants to maximise cooperation. Putting constraints on the clinicians may have caused a lower response rate, resulting in a smaller sample size and less representative sample, introducing bias. In addition, in some cases, placing restrictions on the participants may have introduced unethical intervention in the treatment of the case, for example, in cases in which the horses should have immediate surgery without delay. This limitation may have negatively impacted the performance of the EAAPS.

As opposed to other validation studies [11,26], no control cases were included in the current study since the EAAPS has been validated in previous studies in which colic cases and control cases were included [8,16,17], and the use of the scale in clinical cases of acute colic was the intended purpose of the study. The EAAPS is not intended for use as a diagnostic tool, but rather as a pain severity assessment scale once colic has been diagnosed. Therefore, the use of the scale in controls in real-time was of no significance for the present study. In addition, the prior use of analgesia (by the referring veterinarians) was not recorded along with the study documentation but could have been known by the scorer. Since the purpose of the pain scale is to grade the severity of pain at the moment of evaluation, and since the scores of the scales under comparison were done at the same time, the influence of analgesia was constant between the two scales and, therefore, would not be expected to affect the results. The provision of prior analgesia, however, may have negatively affected the predictive validity, particularly of the EAAPS in comparison to the VAS since the VAS may have taken prior analgesia into account if it were known at the time of assessment.

## 5. Conclusions

In conclusion, the EAAPS was found to be feasible, with a high percentage of participants rating the scoring as easy and quick, particularly when the horse was free to exhibit pain behaviours rather than being in stocks. The advantage of the EAAPS is that it does not require training. The EAAPS was found to be valid in real time and as good as the VAS, except at predicting mortality. In the future, the EAAPS should be tested in the field for use by ambulatory veterinarians as well as by horse riders, caretakers, and farm owners who lack medical training.

## Figures and Tables

**Table 1 animals-10-02242-t001:** The Equine Acute Abdominal Pain Scale. A pain severity score is given based on the score of the behaviour demonstrated. If two or more behaviours are demonstrated, then the score is assigned based on the behaviour with the highest value.

Pain Severity	Behaviours	Score
Mild	No Overt Pain Behaviours	0
↓	Flank Watching Flehman or Lip Curling	1
↓	Sternal Recumbency Stretching Restlessness	2
↓	Kicking Abdomen Pawing	3
↓	Attempting to Lie Down or Crouching Lateral Recumbency	4
↓ Severe	Rolling	5

**Table 2 animals-10-02242-t002:** Characteristics (age and position) of participants (*n* = 35) who scored the pain severity of horses presenting with colic by location and information from the completed forms (*n* = 237) regarding the use of the two scales; Equine Acute Abdominal Pain Scale (EAAPS) and the visual analogue scale (VAS).

Parameter	Category	Israel No. (%)	United Kingdom No. (%)
Age of participants	<25 years	0 (0)	9 (35)
	26–29 years	1 (11)	8 (31)
	30–35 years	1 (11)	5 (19)
	>36	0 (0)	4 (15)
	Missing data	7/9 (78)	0 (0)
	Total	2/9 (22%)	26/26 (100)
Position of participants	Student	5 (56)	NA
	Veterinarian	2 (22)	NA
	Technician	2 (22)	NA
	Total	9/9 (100)	0/25 (0)
Form completion	EAAPS & VAS	18 (95)	108 (50)
	VAS	1 (5)	68 (31)
	EAAPS	0 (0)	39 (18)
	neither	0 (0)	3 (1)
	Total	19/19 (100)	218/218 (100)

NA = not available.

**Table 3 animals-10-02242-t003:** Characteristics of horses presented with colic that were included in the study (*n* = 231) assessing colic pain severity by location (Number (%)).

Variable	Category	Israel (*n* = 13 *)	United Kingdom (*n* = 218)	Total (*n* = 231)
Breed	Cob/Welsh	0 (0)	52 (24)	52 (28)
	Warmblood	0 (0)	39 (18)	39 (21)
	Thoroughbred	0 (0)	23 (11)	23 (13)
	Arab/Arab crosses	6 (46)	9 (4)	15 (8)
	Pony and Miniature	0 (0)	12 (5)	12 (7)
	Others	6 (46)	37 (17)	43 (23)
	Missing	1 (8%)	46 (21%)	47 (20%)
Age	1–5 years	5 (38.5)	29 (13)	34 (15)
	6–10 years	5 (38.5)	76 (35)	81 (35)
	11–15 years	0 (0)	40 (18)	40 (17)
	>16 years	2 (15)	68 (31)	70 (30)
	Missing	1 (8%)	5 (2%)	6 (3%)
Sex	Gelding	1 (8)	129 (59)	130 (56)
	Female	6 (46)	75 (34)	81 (35)
	Female pregnant	3 (23)	5 (2)	8 (3)
	Stallion	2 (15)	5 (2)	7 (3)
	Unknown	1 (8)	4 (2)	5 (2)

* 13 horses contributed 19 forms in Israel.

**Table 4 animals-10-02242-t004:** Predictive Validity: The ability of the visual analogue scale (VAS) and the Equine Acute Abdominal Pain Scale (EAAPS) to predict mortality or surgical treatment in the horses presented with colic. Presented are; the area under the curve (AUC) of receiver operating characteristic curve with 95% confidence intervals (95% CI), the *p*-value and the cut-off values at the highest likelihood ratios of a positive test (LR +).

Parameter	AUC	95% CI	Cut-Off Value	LR +	*p*-Value
Mortality					
EAAPS	0.65	0.54–0.76	4.5	5.5	0.002
VAS	0.82	0.74–0.90	75.5	12.8	<0.001
Surgical Treatment					
EAAPS	0.69	0.60–0.78	3.5	3.3	<0.001
VAS	0.73	0.65–0.82	66.5	3.4	<0.001

**Table 5 animals-10-02242-t005:** The feasibility (quickness and ease-of-use) of the Equine Acute Abdominal Pain Scale (EAAPS) reported as the response of the first-time user by location (Israel; *n* = 9, UK; *n* = 22; UK missing; *n* = 4).

**Quickness**					
**Location**	**Very Quick**	**Quick**	**Within Reasonable Time**	**Slow**	**Very Slow**
Israel	1	3	2	3	0
UK	8	11	2	1	0
Total	9	14	4	4	0
Ease-of-Use					
Location	Very Easy	Easy	Not Easy/Not Difficult	Difficult	Very Difficult
Israel	1	4	2	2	0
UK	4	8	5	5	0
Total	5	12	7	7	0

**Table 6 animals-10-02242-t006:** The feasibility (quickness and ease-of-use) of the Equine Acute Abdominal Pain Scale (EAAPS) reported by first-time users by age group (proportion of responses in each category) (UK data only; *n* = 22; missing; *n* = 4).

**Quickness ^a^**					
**Age Group (Years)**	**Very Quick**	**Quick**	**Reasonable Time**	**Slow**	**Very Slow**
≥ 35 (*n* = 4)	4/4 (100%)				
30-34 (*n* = 4)	3/4 (75%)	1/4 (25%)			
26-29 (*n* = 7)	1/7 (14%)	5/7 (71%)	1 (14%)		
≤ 25 (*n* = 7)	0 (0%)	5/7 (71%)	1 (14%)	1 (14%)	
Ease-of-Use ^b^					
Age Group (Years)	Very Easy	Easy	Not Easy/Not Difficult	Difficult	Very Difficult
≥ 35 (*n* = 4)	2/4 (50%)	1/4 (25%)		1/4 (25%)	
30-34 (*n* = 4)	1/4 (25%)	1/4 (25%)	1/4 (25%)	1/4 (25%)	
26-29 (*n* = 7)	1/7 (14%)	5/7 (71%)	1/7 (14%)		
≤ 25 (*n* = 7)	0 (0%)	1/7 (14%)	3/7 (43%)	3/7 (43%)	

^a^
*p* < 0.01; ^b^
*p* = NS.

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
