# Peer review of "Prospective Feasibility and Revalidation of the Equine Acute Abdominal Pain Scale (EAAPS) in Clinical Cases of Colic in Horses"

_animals, 2020, doi:10.3390/ani10122242_

Round 1

Reviewer 1 Report

Please see attached file!

Reviewer 2 Report

Reviewer comments for Animals-966670

Summary

Includes several overstated claims “proved valid”, “only pain scale that has been tested”, “requires no training”.

Abstract

No explanation about “previously validated” or “revalidated”

No objective measures of the different types of validity listed or how they were determined

There is a described training effect present, despite claims of no training needed

Unclear what types of clinical cases of “colic” were evaluated – medical, surgical, other

Unclear how VAS was used or compared to EAAPS

Introduction

48 - Limit significant digits to clinically relevant levels – 31%

64 – Need more specific details of this prior study as the authors have an evident bias to this work and it forms the foundation of the current work

69 – need details of “simple descriptive scale”.  Not useful to just list a simple phrase

70 – redundant words – reproducibility or repeatability

75 – The term feasibility is introduced and of all the potential characteristics are listed; however, only two were selected for use in the project.  Need to justify.  What is the difference between “quickness” and “ease of use”?  For quickness - Is 1 second (fast) versus 5 seconds (slow) clinically relevant?

84 – Validity cannot be “re-evaluated” as it was not ever tested in this setting before – only presumable from video clips in a prior study.

84 – Descriptions of validity need to be moved up in the introduction as they are mixed in with the objectives and hypotheses.

86 – No description of what was used as a presumed ‘gold standard’

Materials and Methods

94 – Need description on inclusion of three groups of evaluators – assumed be an assessment of ‘experience”?

112 – Not clear how the VAS “biased” the EAAPS when they were both listed on the same form and sued subsequently on repeated cases – ie the examiners would learn to know the measured outcomes of both tools unless it was the very first time using the tools

120 – Severity here is listed 1-5, but described on 11 as (0-5) and used in the study on 110 as (0, A-E)

137 – If the validation were previously described, why are they described again here?  One could argue that the VAS is a poor measure of pain and therefore a poor choice for use as a “gold standard”, if indeed that was done.

Why were these tools not used post-operatively when there is a known source of abdominal pain or at least the medical versus surgical status of the colic was used in determining some sort of validation?

148 – Need justification or supporting evidence for the construct that severe pain ~ higher mortality

159 – Okay – now mentioning first time users and feasibility

161 – More than three time, but how many times?

163 – Unclear what “bias of clustering” refers to and how this was limited

164 – So they scored the EAAPS twice – with the second time in reverse order of severity?  Unclear what and why this was done

177 – This is the first mention of mortality or surgical treatment.  These need to be described in detail in the materials and methods.  Not all medical treatments are equal.  Not all surgical cases are equal – obviously a simple displacement is different from a mild to severe torsion or a small or large resection.  Not all mortalities are equal – some are based on financial concerns or co-morbidities and horses are euthanized before any surgery is considered.

187 – First mention of “different age groups”.  Age does not equate to experience unless the number of years of experience are considered

Results

192 – So five horses had recurrent colic episodes and were scored twice?

196 – Unclear how heart rate was used in the analysis as it was not mentioned

204 – Age is not really useful information.  Experience is needed.

Table 1 – Age - Error of 26/26 = 100%, should be 26/28

Form completion error – 19/237 and 218/237

Table 2 – Total numbers (231) do not compare to numbers in table 1 (237).  Explain

Unclear what 12/13 and 172/218 refers to

Age categories not defined in methods

Why are pregnant mares lists separately as a different sex or gender?

230 – So how was “mortality” judged to be a reflection of “pain severity” as proposed as a construct?  How was this adjusted for to match the reality of surgery and surgical pain as proposed in your construct?

238 – Definitions need to be provided a priori for the subjective score of “substantial” and “good”.

241 – No methods or measures of “association” were listed in the analysis.

Table 4 – Not clear how user location is useful information.  Was not listed in the methods

262 – Redundant information - how does “experience” relate to “learning curve”?

Table 6 – Still unclear how this was calculated or how useful the information is

285 – Not useful details

291 – While useful, not included in the methods or how used to improve the pain assessment tool

299 – Overstated – does this include humans and dogs too?

301 – Overly broad use of the term “feasibility”

311 – Overstated - Reliability was not assessed in this study as there was no proven gold standard

318 – First mention of some unit of time (matter of minutes) needed to use the tool.

322 – Obvious knowledge of colic behaviors is needed unless tested on non-veterinary population

340 – Overstated claims of superiority over undefined tools

376 – Assumes that the reader knows of version 1 and 2.

392 – So why was heart rate recorded?

Very long and redundant discussion

Conclusions

First mention of free versus stocks assessment

435 – How many times do you have to say that it was quick and easy to use?

435 – So why not just use the VAS?

Supplemental

Major limitation - No descriptions of when and if prior analgesic were given relative to the scoring

Figure S2b – Do not see the value of this figure

Reviewer 3 Report

Dear, please the description of the figure 1 should be stated below!

Author Response

Reviewer comment – The description of figures should be stated below!

Author response – The description of the figure was moved to below the figure.

Reviewer 4 Report

In this paper, the Authors evaluate the use of EAAPS in horses with colic at two

equine hospital. 

The Authors have investigated an interesting topic.

The manuscript is well written, presented and discussed.

In general, the structure of the article is satisfactory and in agreement with the journal instructions for authors.

The objectives of the paper are of interest and fit well within the scope of the journal.

In my opinion, the manuscript could be accepted for publication in Animals.

Author Response

Reviewer comment – General comments

In this paper, the Authors evaluate the use of EAAPS in horses with colic at two equine hospital.

The Authors have investigated an interesting topic.

The manuscript is well written, presented and discussed.

In general, the structure of the article is satisfactory and in agreement with the journal instructions for authors.

The objectives of the paper are of interest and fit well within the scope of the journal.

In my opinion, the manuscript could be accepted for publication in Animals.

Author response – Thank you.

Reviewer 5 Report

The authors have prepared a well-written manuscript about an important topic - assessment of equine colic pain. Thus, the manuscript is worthy of publication in Animals. Although the Equine Acute Abdominal Pain Scale (EAAPS) seems to be relatively easy to use, its impact in a clinical setting would like be more as a learning tool for veterinary students than as a decision making aide for experienced clinicians. Further, a limitation of the study arising in the written comments that needs to be acknowledged as a significant limitation of the study was that horses appear to have been assessed in various settings that are not well described (e.g., retrained in stocks, loose in a stall, etc.). Horses unrestarined in a stall are commonly recognized to show more signs of colic pain than when restrained with a lead rope or in stocks. It is unfortunate that the EAAPS, as currently described in the manuscript, was not performed in a more standardized setting although it is recognized that it would be unwise to place a painful horse in a stall for performance of the EAAPS rather than to proceed directly to surgery or euthanasia. Next, heart rate was reported but there was not attempt to correlate heart rate to EAAPS or the visual analog scale (VAS). Traditionally, heart rate has been documented to be a fairly good prognostic measure for colic outcome (but perhaps not pain severity). Heart rate could be a potential "gold standard" for comparison for the pain scales in this study - significant correlations between heart rate and both the EAAPS and VAS could be expected and including these comparisons would strengthen this manuscript. Finally, I encourage the authors to replace "depression" (defined as a "feeling" sensed by an individual) to either "lethargy" or "dull attitude" as we cannot truly assess depression in horses.

There are a few additional minor comments/corrections for the authors to consider:

Lines 68, 127, and 170 - "scale" can be deleted after EAAPS a the S is for "scale"

Line 150 - the ";" after "as" seems unnecessary

Line 175 - "score" can be deleted after EAAPS

Line 180 - "statistically" can be deleted

Line 218 - I suggest starting with the number of horses in the study here "Of the 231 horses included in the study, the most common breed ..."

Table 2 - I am struggling to understand why the values for breeds presented do not sum to 100%? Should there be an additional line for "not reported"? 

Line 241 - positive test of 6.4 - not sure what this means? either mortality or surgical intervention - please clarify as the value 0.64 does not appear in Table 3.

Tables 4 and 5 - it is unclear why certain values are inconsistent: 1) Table 4 - UK quick = 12 but Table 5 - UK quick = 11; 2) Table 4 - UK very easy = 3 but Table 5 - UK very easy = 4; 3) Table 4 - UK easy = 9 but Table 5 - UK easy = 8 - please double check and either correct values or explain the inconsistencies

Table 6 - probably not necessary, consider deleting

Lines 385-387 - please restructure this sentence - a bit long and confusing as currently written 

Lines 389-393 - the authors could provide support for heart rate not being a good assessment of pain if they found poor correlations between heart rate and the pain scales used.

Lines 433-434 - this appears to be the only mention of seeing horses free (assume loose in a stall) as compared to being in stocks; this information needs to be clearly presented in the Methods (and perhaps the Results)
